# Factors influencing participant compliance in acupuncture trials: An in-depth interview study

**Hui-Juan Cao**[1]*, **Xun Li**[1], **Xin-Lin Li**[2], **Lesley Ward**[3,4], **Zhan-Guo Xie**[5], **Hui Hu**[6], **Ya-Jing Zhang**[7], **Jian-Ping Liu**[1]*

**1** Centre for Evidence-Based Chinese Medicine, Beijing University of Chinese Medicine, Beijing, China, **2** China Press of Traditional Chinese Medicine, Beijing, China, **3** Department of Sport, Exercise & Rehabilitation, Northumbria University, Newcastle upon Tyne, United Kingdom, **4** Australian Research Centre in Complementary and Integrative Medicine, University of Technology Sydney, Ultimo, Australia, **5** The Department of Acupuncture and Moxibustion, First People's Hospital of Dongcheng District, Beijing, China, **6** The Department of Acupuncture and Metabolic Diseases, Dongfang Hospital Affiliated to Beijing University of Chinese Medicine, Beijing, China, **7** Institute of information and literature, Jiangxi Institute of Traditional Chinese Medicine, Jiangxi Province, China

☯ These authors contributed equally to this work.
* huijuancao327@hotmail.com (HJC); jianping_l@hotmail.com (JPL)

**Data Availability Statement:** Data cannot be shared publicly because of recordings of the conversations are private, which contain potentially identifying and personal information. Data are available from the Ethics Committee of Dong Zhi

## Abstract

### Introduction

Little is known of acupuncture patients' experiences and opinions of clinical trials, and what may influence their compliance when participating in an acupuncture trial.

### Objectives

To explore the potential factors that influence patients' choice and determinants to participate in acupuncture clinical trials.

### Methods

Ten qualitative, in-depth interviews were conducted with patients from acupuncture clinics in Beijing, who had previously participated in acupuncture clinical trials.

### Results

Four main themes emerged from the interview data: effectiveness of the treatment, convenience of participating in a trial, doctor-participant communication, and participant acceptance of the treatment (or the trial). Effectiveness of acupuncture in treating the health condition was the most important factor for participant adherence. Pragmatics of treatment schedules, travel and attendance burden, together with confidence in the doctor's ability additionally influenced trial and treatment compliance.

Men Hospital (contact via Dr. Yu) for researchers who meet the criteria for access to confidential data. The data underlying the results presented in the study are available from Dr. Changhe Yu (yakno2@163.com), via below mail address: Department of massage, Dong Zhi Men Hospital, No. 5 of Haiyuncang Road, Dongcheng District, Beijing, 100701 China.

**Funding:** Hui-Juan Cao is supported by the National Natural Science Foundation of China (81804000) and the Beijing Municipal Organization Department (2017000020124G292). Jian-Ping Liu is supported by the Key project of the National Natural Science Foundation of China (No.81830115). The funders had no role in study design, data collection and analysis, decision to publish, or preparation of the manuscript.

**Competing interests:** The authors have declared that no competing interests exist.

## Conclusions

In-depth interviews suggest that treatment effectiveness, the pragmatics of attending treatment sessions, and the expertise and attitudes of acupuncturists are determining factors of participation and compliance in acupuncture clinical trials. Participants' confidence in, and expectation of, acupuncture may facilitate compliance, while their fear of acupuncture and negative perceptions of the trial's purpose may reduce treatment compliance. Compliance may be facilitated by enhanced doctor-patient communication, personalized treatment programs, and feedback on treatment outcomes.

## Background

Randomized controlled trials (RCTs) are established as the gold standard to determine intervention efficacy, due to their rigorous design. However, the delivery of an intervention within the structure of an RCT may require participants to follow a sometimes restrictive treatment strategy, which may be contrary to their personal treatment intentions. As a non-drug therapy, the degree of acceptance of acupuncture therapy may be affected by patients' values and preferences. If the patients were randomly assigned to an acupuncture group but did not prefer the treatment regime, this may lead to a higher rate of trial withdrawal or intervention attrition.

Attrition and poor compliance are common in acupuncture trials, and the resulting missing data has a serious impact on the robustness of study results. As reported in our previous literature review, only one third of published acupuncture trials in China reported no participant withdrawal or missing follow-up data. Our re-analysis of data from papers included in the review indicated that 85.2% of the studies may have reported false positive results by not taking into account missing data [1]. To improve compliance, it is necessary to understand from the participants' perspective what factors may positively or negatively influence their adherence to both the trial and to the intervention treatments.

Previous research has investigated potential factors influencing trial recruitment and participant compliance [2–4]. These studies found no correlation between participant characteristics, study setting, funding sources, recruitment rates, or attrition. Acupuncturists have proposed factors that may improve participant compliance in acupuncture RCTs, based on their own experience [5, 6]. These factors included fully informed consent, improvement of medical environments such as cleanliness of the treatment facility and appointment waiting times, and good doctor-patient communication. However, as these factors are all from the perspective of the researcher or acupuncturist, they do not necessarily represent participants' feelings or attitudes.

Our previous cross-sectional survey, based on 500 respondents, suggested that completion of the acupuncture treatment within the specific time regimen (relating to frequency and number of treatment sessions) maybe the primary factor influencing treatment compliance [7]. However, as only 8% of the surveyed participants had previously participated in the RCTs the survey results were more related to factors influencing potential participation in clinical research. As such, a lack of information remains regarding factors influencing treatment of compliance.

To address this knowledge gap, the aim of the current study was to explore the potential factors influencing a patients' choice to participant in, and adhere to, acupuncture RCTs.

## Methods

For the purpose of this study, the term 'patient' refers to an individual considering, but not currently taking part in a trial, while the term 'participant' refers to an individual participating in a trial.

### Study design

We conducted individual, in-depth interviews with 10 people who had previously participated in three separate acupuncture RCTs within the past 5 years. Interviews were conducted over a four-month period, from April to July 2016. This study was approved by the Ethics Committee of Beijing University of Chinese Medicine (Approval number: 2017BZHYLL0316); and is reported according to the consolidated criteria for reporting qualitative health research checklist (COREQ, see S1 Table) [8].

### Participant selection

Participants were selected from the 40 respondents in our previous cross-sectional study [7] who had indicated experience of participation in RCTs. The respondents were from the outpatient Department of Acupuncture and Moxibustion of Dong Fang Hospital, the Department of Endocrine and Metabolic disorders of Dong Fang Hospital, and the outpatient Department of Acupuncture and Moxibustion of the First People's Hospital of Dongcheng District, Beijing, China.

From this convenience sample of 40 potential participants, individuals who met the following criteria were contacted by telephone to take part in the interview study: minimum age 18 years old, previously participated in clinical trials, had no mental illness impacting ability to take part in interviews, and were able to provide informed consent to be interviewed. Potential participants were informed about the researchers' background, occupation and credentials. They were given an information sheet which described the purpose of the research, and advised by the researcher that they could withdraw from the study at any time, and request their audio recordings to be destroyed. Those in agreement then signed a written consent form prior to their interview taking place. Participants were continuously sampled from the convenience sample, with sample size determined according to the principle of information saturation. Recruitment was terminated when no new themes were derived from the interview data.

### Setting

Each interview was conducted in a quiet location of the participant's choice. Nine of the ten interviews were conducted in a doctor's office, and one interview conducted via a video call, for their convenience. Interviews lasted, on average, 30 minutes.

### Data collection

Interviews were conducted according to a *pro-forma* schedule of questions (S2 Table). This schedule was based on the research literature [2–5], results of our previous survey [7], and pre-interview discussions among the research team. The interviewers were two female postgraduate students (BSc.), both majoring in Evidence-based Medicine and trained in qualitative research (XLL and YJZ). They were further trained in conducting the interviews by a member of the research staff with qualitative experience (XL). The interviewers had no connection with the participants prior to this study. All interviews were audio taped, and transcribed verbatim;

then independently checked for accuracy against the audio by a member of the research team (ZGX).

## Data analysis

The two interviewers (XLL and YJZ) transcribed the recordings verbatim, including the indication of non-verbal responses such as sighs, interjections, and facial expressions. Following transcription, two researchers (XLL and HJC) independently coded the transcription data, discussed and merged codes after each interview, and consulted another researcher (JPL) if they could not reach consensus. HJC and XL sorted the preliminary findings into sub-themes and themes. Data analysis was iterative, and ongoing through the course of the study, with the codes, sub-themes, and themes constantly compared with the transcript data. The research team (HJC, XL, XLL and JPL) discussed all codes, sub-themes and themes after saturation, and constructed the final thematic map.

# Results

## Description of the participants

In total, 10 of the 40 people contacted agreed to take part (25% recruitment rate), and all participants completed the study (0% attrition). When the analysis of the 9th and 10th interview produced no new codes or concepts, we determined that data saturation had been reached, and no further participants were recruited.

Participant characteristics are shown in Table 1. Participants were aged between 27 to 65 years old, and all had fibromyalgia. Of the 10 participants, eight were female, and seven were from the local Beijing area. Four participants were employed, and six were unemployed or retired. Two participants had to pay for their own treatment, while the other eight people had medical insurance. All had previously participated in one of three acupuncture RCTs conducted at two hospital sites, and were assigned anonymous identification numbers based on their hospital site and order of selection.

## Thematic overview

A finalized map of codes, sub-themes, and themes is presented in Table 2. Four main themes were derived from the interview data, representing key factors influencing a participant's

**Table 1. Participant characteristics.**

| No. of organization | No. of interviewee | Gender | Age | Location of household registration | Occupation | Payment method |
|---|---|---|---|---|---|---|
| 1 | 1.1 | Male | 65 | Beijing | Retired | Medical insurance |
| | 1.2 | Female | 35 | Beijing | Unemployed | Medical insurance |
| | 1.3 | Female | 51 | Henan | Unemployed | Non-insurance |
| | 1.4 | Male | 36 | Beijing | Employed | Medical insurance |
| | 1.5 | Female | 54 | Beijing | Retired | Medical insurance |
| | 1.6 | Female | 53 | Beijing | Unemployed | Medical insurance |
| 2 | 2.1 | Female | 45 | Inner Mongolia | Employed | Medical insurance |
| | 2.2 | Female | 37 | Beijing | Employed | Medical insurance |
| 3 | 3.1 | Female | 27 | Hebei | Employed | Non-insurance |
| | 3.2 | Female | 51 | Beijing | Retired | Medical insurance |

1. Department of Acupuncture and Moxibustion, Dong Fang Hospital

2. Department of Endocrine and Metabolic Disorders, Dong Fang Hospital

3. Department of Acupuncture and Moxibusion, the First People's Hospital of Dongcheng District

**Table 2. Final thematic map.**

| Theme | Sub-theme | Code | Illustrative quote |
|---|---|---|---|
| Effectiveness of the treatment | Original intention of participating in the trial | The illness persisted | "It has been a long time. . .feel really painful, and I used a lot of treatments which did not work. I just want to try something else." (participant 1.2) |
| | | Other therapies are ineffective | "I don't know about it (clinical trial). Because the drugs were not work, there is no other good choice, so I just want to have a try (of acupuncture), that's what I thought." (participant 1.1) |
| | | Attempt to use acupuncture | |
| | | Hope for relief of symptoms | "I do not understand this (clinical trial). . . .I just want to cure it (the pain)." (participant 1.3) |
| | Reasons for completing the trial | Previous acupuncture therapy was effective | "Um. . . . (I'm satisfied with) the results, yes. The first time (I was treated with acupuncture) I thought it was amazing." (participant 3.1) |
| | | The current acupuncture treatment is effective | "The pain has eased in all these places." (participant 1.5) |
| | | Doctor's skill is the most important factor | "If a good doctor gives the (acupuncture) treatment, the effect will certainly be better." (participant 1.2) |
| | | Want to verify whether acupuncture is effective | "The important thing is, is to stick to it. It must be useless if you only take acupuncture once or twice." (participant 1.2) |
| | Effectiveness influences decision-making | Ask for extending the trial's period when it works | "Yeah, it's almost there (cured). But I think it should be more (treatment sessions) . . .I want to join another round, to consolidate the results." (participant 1.6) |
| | | Use other alternatives if it does not work well | "I'm not keep using one treatment. If I take drugs here (in this hospital) for 7 or 10 days, and it does not work, I will change another (therapy)." (participant 1.3) |
| Convenience of participating in the trial | Treatment schedule | Treatment frequency | "There is no difficulty, . . . but (I can come) at most three times a week. Many people want to come, but this time is hard for them, because the working people cannot guarantee to be here three times a week. So, they can't participate in this, only us retired elder people can guarantee to come. Some of my neighborhoods also suffer from back pain. They are in their 50s, but they are all at work now. You can't let him come three times a week." (participant 3.2) |
| | | Waiting time before treatment | |
| | | Conflict with working hours | |
| | Distance from hospital | Distance from home to hospital | "Sure, I will recommend my friends to join in the trial, (pause) if they live nearby. They won't come if it is too far." (participant 2.2) |
| | | Distance from work place to hospital | "I sit in the office, when I remembered, when there was nothing else to do, or when I had time I came here. My workplace was close to here, it won't take too much time walking." (participant 1.4) |
| | | Frequent visitor of the hospital | "I'm treated here (this hospital) every week, so it's very convenience to attend this (trial)." (participant 1.6) |
| Doctor-participant communication | Purpose of the communication | Help doctor to precisely identify the lesion location | "Without the communication, doctors cannot find the best treatment for your disease. Communication, fully communication is necessary. Otherwise, it not works, and you may not feel at ease." (participant 1.1) |
| | | Relieve the participant's tension/ nervousness | "It (communication) also made me feel better, so that I would not be particularly afraid, would not feel that the hospital is a particularly terrible place." (participant 3.1) |
| | | Enhance participant's confidence of healing the disease | "One thing is that the patient needs the encouragement and affirmation from the doctor. If the doctor says that 'the disease is all right, if you cooperate with the treatment well, we have confidence to ease it for you'. Then the patient feels confident and feels that it is meaningful to insist on the treatment. . . . Just like educating a child, if you always scold him for being a fool, he will lose confidence. So, patients also need doctors to give him confidence, . . . if there is such communication, patients will be particularly confident, no less than taking a dose of decoction." (participant 3.2) |
| | Expectations for doctor's attitude | Good communication reflects doctor's sense of responsibility | "If the doctor is willing to communicate with the patient, I think he is a conscientious and responsible doctor, I would like to always find him to help me with treatment. If that kind of doctor, who just do the acupuncture without ask patients about the condition, I think this is not a responsible doctor." (participant 2.2) |
| | | Being remembered by doctors makes a patient feel good | "Every time she (the acupuncturist) can recognize me, I think it's a good feeling." (participant 1.6) |
| | | Hope to get feedback after the treatment | "I still hope to have a feedback, I want to know, um, I'm not particularly sensitive to changes in my body, I hope to get more professional explanations to let me know about changes in my body, hope the doctor from a professional point of view to give me an answer." (participant 3.1) |

*(Continued)*

**Table 2.** (Continued)

| Theme | Sub-theme | Code | Illustrative quote |
|---|---|---|---|
| Participant's acceptance of the treatment (or trial) | Participant's belief | Acupuncture is a traditional therapy | *"I think it's thousands of years' history of acupuncture and moxibustion in China. It's something of our ancestors that will work."* (participant 1.1) |
| | | Side effects of acupuncture are less than drugs | *"I think acupuncture. . .um. . . is a traditional (treatment), . . . unlike some drugs, (some drug) is completely new (therapy). . . . There's no risk (of the acupuncture treatment). It is traditional, I believe it."* (participant 2.1) |
| | | Regular hospital treatment is reassuring | *"Because (this hospital) is a regular hospital, not a private one, . . ."* (participant 1.6) |
| | Participant's worries | Pain caused by the treatment itself | *"Because this is my first time of acupuncture, I'm afraid to be hurt."* (participant 3.1) |
| | | It's painful to hold a position for a long-time during acupuncture | *"I'm just afraid of the pain, there's nothing else to worry about."* (participant 1.5) |
| | | Be treated as a laboratory rat in the trial | *"(I was) worried to be treated as laboratory rat. . . . If they (the acupuncturists) had bad skills, it (acupuncture) would be painful."* (participant 2.2) |
| | Potential benefits from the trial | Free treatment | *"We also have other basic diseases, chronic diseases. We have to take medicine for a long time. The cost is high. So, the budget for massage and acupuncture is very small."* (participant 3.2) |
| | | High-quality treatment | *"(In) This experiment (they) will arrange the chief to do the acupuncture to teach the students. I think this opportunity is very rare."* (participant 2.2) |
| | | Providing data for scientific research | *"It is also to provide a basis for scientific research, . . ., after all, clinical research can only be developed on the basis of it (research)."* (participant 2.2) |

adherence when participating in an acupuncture trial: 1) effectiveness of the treatment; 2) convenience of participating in the trial; 3) doctor-patient communication; and 4), and patient's acceptance of the treatment (or the trial).

**Theme 1-Effectiveness of the treatment.** Almost all interviewees stated that the effectiveness of the acupuncture treatment delivered in the clinical trial was an important factor influencing their trial adherence. Treatment effectiveness was considered dependent on both whether the treatment plan aligned with their condition, and the expertise of the acupuncturists. All interviewees stated their target outcome for participating in an acupuncture trial was pain resolution, with the therapeutic effect of acupuncture the primary reason for adhering to trial completion. Three sub-themes were associated with treatment effectiveness, as follows.

*Subtheme 1.1 Original intention of participating in the trial.* Most interviewees knew very little about clinical trials. Their first concern was whether their condition met the recruitment criteria for the trial, which in turn would impact whether the experimental treatment may have a therapeutic effect on their condition. Some interviewees had suffered from chronic pain for several years, and tried numerous ineffective therapies. As such, they were seeking potential effect treatment through joining a trial:

> *"I don't know about it (clinical trial). Because the drugs were not work, there is no other good choice, so I just want to have a try (of acupuncture), that's what I thought."* (participant 1.1)

*Subtheme 1.2 Reasons for completing the trial.* All interviewees expressed satisfaction with the acupuncture received during their trial participation, and reported their symptoms had improved to varying degrees as a result of the treatment. This outcome encouraged them to fully adhere to the treatment schedules. Additionally, some mentioned that they wanted to complete the trial to verify the effect of acupuncture, as only completing part of the treatment may have been ineffective:

*"The important thing is, is to stick to it. It must be useless if you only take acupuncture once or twice."* (participant 1.2)

*Subtheme 1.3 Effectiveness influences decision-making.* While all 10 interviewees had completed their respective trial treatments, some indicated they had thought of stopping earlier if the treatment was not effective:

*"I'm not keep using one treatment. If I take drugs here (in this hospital) for 7 or 10 days, and it does not work, I will change another (therapy)."* (participant 1.3)

Those who felt benefits also indicated a preference to continue with treatment, to further improve their condition:

*"Yeah, it's almost there (cured). But I think it should be more (treatment sessions) . . .I want to join another round, to consolidate the results."* (participant 1.6)

**Theme 2-Convenience of participating in the trial.** The convenience of participating in a trial was determined by the trial's response burden: the balance between the time and energy expended by the participants versus the benefits of the trial treatment. This burden was unique to each individual. For example, for someone who was retired, the distance to the hospital for treatment may be of primary consideration, while for an employed person the waiting time and frequency of treatment are more important. Trial convenience was grouped in two main areas, as outlined below.

*Subtheme 2.1 Treatment schedule.* Similar to our previous survey's main finding [7], most interviewees stated they would hesitate to join in a trial if the treatment schedule was too frequent. Attending a hospital for acupuncture treatment more than twice a week was considered difficult for those who were currently employed, compared to those who were retired:

*"There is no difficulty, . . . but (I can come) at most three times a week. Many people want to come, but this time is hard for them, because the working people cannot guarantee to be here three times a week. So, they can't participate in this, only us retired elder people can guarantee to come. Some of my neighborhoods also suffer from back pain. They are in their 50s, but they are all at work now. You can't let him come three times a week."* (participant 3.2)

*Subtheme 2.2—Distance from the hospital.* In addition to treatment frequency, the amount of travel required to attend a treatment was also a concern. Interviewees living or working close to a treatment centre stated they were more likely to participate in a clinical trial:

*"Sure, I will recommend my friends to join in the trial, (pause) if they live nearby. They won't come if it is too far."* (participant 2.2)

**Theme 3-Doctor-patient communication.** Interviewees thought that the quality of doctor-patient communication was dependent on both the patient's confidence and the doctor's attitude. Patients need to be able to express their illness as completely and in as much detail as possible, while doctors need to be willing to adopt a friendly attitude to listening and communication. Two subthemes were association with this communication, as below.

*Subtheme 3.1 Purpose of communication.* Most interviewees thought that doctor-patient communication was very important, for the following reasons. Through good doctor-patient communication, doctors can understand the patients' condition in detail, to better formulate

accurate acupoint selection. This in turn increases trust between doctors and patients, increases patients' confidence in their treatment, and reduces any psychological pressure, such as anxiety, associated with their treatment. Collectively, interviewees associated these factors with achieving a better therapeutic effect:

> *"Without the communication, doctors cannot find the best treatment for your disease. Communication, fully communication is necessary. Otherwise, it not works, and you may not feel at ease."* (participant 1.1)

*Subtheme 3.2 Expectation for doctor's attitude.* Interviewees viewed the doctor's attitude as reflecting their sense of responsibility, with a responsible doctor taking time with a patient to explain their condition and listen to their concerns. Interviewees also stated they wanted doctors to provide feedback on treatment outcomes, and provide them with the support to better understand their condition:

> *"One thing is that the patient needs the encouragement and affirmation from the doctor. If the doctor says that 'the disease is all right, if you cooperate with the treatment well, we have confidence to ease it for you'. Then the patient feels confident and feels that it is meaningful to insist on the treatment. . . . Just like educating a child, if you always scold him for being a fool, he will lose confidence. So, patients also need doctors to give him confidence, . . . if there is such communication, patients will be particularly confident, no less than taking a dose of decoction."* (participant 3.2)

**Theme 4-Patient's acceptance of the treatment (or trial).** In clinical practice, many patients may not be familiar with acupuncture, and most will have limited experience of participating in clinical trials. As such, several factors may affect their acceptance of receiving acupuncture treatment or participating in a trial itself. Patients who believe in traditional Chinese medicine may be more willing to accept acupuncture therapy and participate in trials. Conversely, those who fear acupuncture therapy, or who believe that clinical trials are a form of experimenting on them may be more reluctant to participate in trials. Three sub-themes relate to this issue.

*Subtheme 4.1- Patient's belief.* The interviewees, who had all previously participated in acupuncture trials, were notably convinced of the use of Chinese medicine. Most of them viewed acupuncture as a traditional treatment, with its history of thousands of years of practice indicative of its enduring effectiveness. Moreover, interviewees were assured by the fact the acupuncture clinical trials were conducted in regular hospitals. They also believed that acupuncture had fewer side effects than drugs:

> *"I think acupuncture. . .um. . . is a traditional (treatment), . . . unlike some drugs, (some drug) is completely new (therapy). . . . There's no risk (of the acupuncture treatment). It is traditional, I believe it."* (participant 2.1)

*Subtheme 4.2 Patient's worries.* Most interviewees had no experience of acupuncture before they took part in their respective clinical trials. Their primary concerns coming into the studies were potential pain, inflammation or side effects caused by needling; with fear of pain still present even after receiving the acupuncture. Other concerns included the high cost of treatment, and the level of qualification of acupuncturists involved in clinical trials, with concern that they would be experimented on like lab animals:

> *"(I was) worried to be treated as laboratory rat. . . . If they (the acupuncturists) had bad skills, it (acupuncture) would be painful."* (participant 2.2)

**Subtheme 4.3—Potential benefits from the trial.**   Some interviewees were frequent patients at the trial hospitals, which carried a heavy financial burden. As such, they were happy that participation in clinical trials presented a means of free treatment. Additionally, trial participation was seen as an opportunity to be treated by a highly qualified doctor:

> *"(In) This experiment (they) will arrange the chief to do the acupuncture to teach the students. I think this opportunity is very rare."* (participant 2.2)

Some interviewees also cited their willingness to take part in clinical trials as a means to facilitate scientific research, and provide a scientific basis for the promotion of acupuncture in the future.

## Discussion

### Summary of main findings

In-depth interviews of ten individuals who had previously participated in a range of acupuncture RCTs indicated four common themes relevant to trial compliance: effectiveness of the treatment, convenience of participating in a trial, doctor-participant communication, and participant's acceptance of the treatment (or the trial). Of primary importance for compliance was effectiveness of the acupuncture treatment. Pragmatic issues, such as timing of the sessions, response cost of attending the treatments, and quality of the doctor-patient communication additionally affected trial and treatment compliance.

### Comparison with existing literature

Our study both supports and furthers previous research findings. In those studies where secondary analysis of clinical trial data was performed, we confirm previous finding of a relationship between treatment effectiveness and participant compliance [9]. We confirmed the importance of doctor-participant communication in trial and treatment compliance; and have additionally presented the perspective of the participants' expectation of the doctor's attitude to their treatment.

This study confirms findings from our previous survey [7], of the importance of the treatment effectiveness to compliance. In addition, the current study also found the response cost of trial participation may influence the decision to continue with treatment. Aspects of response cost highlighted by the interviewees included time spent attending and receiving treatment, as determined by the frequency and number of sessions. According to the results of a UK national survey in 2012 [10], 72.56% of males and 76.99% of females seeking acupuncture treatment are 24–64 years old, with half under 45 years old. Since current acupuncture treatment duration in trials varies from once daily to at least twice weekly excluding weekend days [11], it may cause difficulties for employed participants to attend treatment appointments during their working hours.

Further to previous studies, the current study also found that some of the concerns and expectations of patients participating in RCTs were related to patients' acceptance of the acupuncture trial. It suggested that we should consider the pain sensation caused by acupuncture operation itself when studying patients' compliance of the acupuncture trial. Previous studies [12, 13] have already found acupuncture perception had a tendency to be rated higher when

needling inserted deep; thus, there may be a relation between the depth of needle insertion and perceived effectiveness of the acupuncture treatment [14, 15]. As such, depth of needling should be considered as a factor in the design of future acupuncture research.

## Strengths and limitations

Few studies have specifically addressed the issue of participant compliance in acupuncture clinical trials. Previous studies focused on compliance have mostly analyzed potential influencing factors through statistical results of RCTs [16], or in the form of retrospective studies [17]. In the current study, in-depth interviews with robust qualitative methodology were used to explore potential factors affecting compliance from the subjective perspective of participants for the first time, with this form of in-depth analysis both confirming past findings, and providing new insight into the important factors influencing treatment compliance.

A limitation of our study was targeting participants who had previously achieved good compliance in their respective trials, as their opinions may not fully reflect those of participants who declined or withdrew from study participation. However, as the purpose of the current qualitative study was to better understand actual factors that influenced a person's decision as to why they decided to take part and continue in an acupuncture trial, we would not have captured these viewpoints from those who had declined to participate in the original trials. Additionally, given our interviewees had been involved in three different clinical trials, the commonality of themes suggests our results may have relevance to a broader clinical trial population.

A further limitation, inherent to qualitative research, is that our findings relating to treatment effectiveness, the pragmatics of attending treatment sessions, and the expertise and attitudes of acupuncturists cannot be generalized outside of our study population. However, while there may be differences between China and other countries regarding patients' attitudes to these issues, we believe our underlying results highlight generic trial recruitment and compliance issues, and will provide useful information for researchers to improve these study issues, regardless of country of intervention delivery.

## Implications for future research

We suggest consideration of the above four main themes when designing future acupuncture trials, to improve participants' compliance. Reducing participant response cost could be addressed by developing individualized treatment schedules according to each participant's situation. For example, treatment sessions could be coordinated with pre-planned hospital visits, and the times and frequency of treatment sessions coordinated to be compatible with a participant's work schedule. Additionally, enhanced doctor-patient communication may allay participant concerns, and improve their confidence in the treatment program. As interviewees indicated a lack of prior knowledge of clinical trials, we suggest that future studies should enhance the publicity of trial information. Posters and official Wechat Subscription could be used to publicly and freely disseminate detailed trial information.

To further enhance the knowledge base on participant compliance, we suggest interviews with both participants who withdrew from clinical trials prior to completion, and the acupuncturists or researchers directly involved in the trials, to better understand the factors influencing patient compliance from the perspective of all trial stakeholders, to further improve the quality of acupuncture clinical trials.

## Conclusions

In-depth interviews found that treatment effectiveness, the pragmatics of attending treatment sessions, and the expertise and attitudes of acupuncturists are determining factors of

participation and compliance in acupuncture RCTs. Additionally, confidence in, and expectation of, acupuncture may enhance treatment compliance. Conversely, fear of acupuncture and negative perceptions of research as experiments may be barriers to compliance. To increase participant compliance in acupuncture trials it is suggested that future studies facilitate clear, open doctor-patient communication, consider personalized treatment programs, and enhance follow-up and feedback on treatment outcomes.

## Supporting information

**S1 Table. Consolidated criteria for reporting qualitative research (COREQ): A 32-item checklist for interviews and focus groups.**
(DOCX)

**S2 Table. The outline of the interview.**
(DOCX)

## Acknowledgments

We thank all the participants for their contribution.

## Author Contributions

**Conceptualization:** Hui-Juan Cao, Jian-Ping Liu.

**Data curation:** Xun Li, Xin-Lin Li, Zhan-Guo Xie, Hui Hu, Ya-Jing Zhang.

**Formal analysis:** Hui-Juan Cao, Xin-Lin Li, Lesley Ward.

**Funding acquisition:** Hui-Juan Cao.

**Investigation:** Xin-Lin Li, Zhan-Guo Xie, Hui Hu, Ya-Jing Zhang.

**Methodology:** Hui-Juan Cao, Xun Li, Lesley Ward, Jian-Ping Liu.

**Project administration:** Hui-Juan Cao.

**Supervision:** Hui Hu.

**Writing – original draft:** Hui-Juan Cao.

**Writing – review & editing:** Hui-Juan Cao, Xun Li, Xin-Lin Li, Lesley Ward, Jian-Ping Liu.

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
