## [Decision Letter · Decision Letter 0]

6 Jan 2020

PONE-D-19-30876

Factors influencing participant compliance in acupuncture trials: An in-depth interview study

PLOS ONE

Dear Dr. Cao,

Thank you for submitting your manuscript to PLOS ONE. After careful consideration, we feel that it has merit but does not fully meet PLOS ONE’s publication criteria as it currently stands. Therefore, we invite you to submit a revised version of the manuscript that addresses the points raised during the review process.

In revising your manuscript please address these additional comments:

1. Provide further background on compliance to acupuncture trial among Chinese and other races and why this is relevant in medicine

2. Elaborate further on measures that were taken to reduce selection bias through refusals in this study. Given that this study set out to understand compliance related perspectives of clinical trial patients, by including only those who had good compliance and had repeatedly participated in other trials in the past, has several implications for this study. The issue of non-compliance may not have been adequately explored. Authors should comment on these in the discussion adequately.

3. Elaborate on whether saturation was adequately achieved in this study. 30 of the 40 approached had refused participation. Were there other eligible patients that the study team could have approached had saturation not been reached. The 10 participants also seem to be quite heterogenous in their backgrounds. It is also not explained what type of trials they had participated in for example, whether there were of long/ short duration, included invasive procedures or were meant for specific health conditions. All these will have bearings on the qualitative content. Authors should elaborate on these points in the context of data / thematic saturation and implications of the findings.

We would appreciate receiving your revised manuscript by Feb 20 2020 11:59PM. To enhance the reproducibility of your results, we recommend that if applicable you deposit your laboratory protocols in protocols.io, where a protocol can be assigned its own identifier (DOI) such that it can be cited independently in the future. For instructions see: http://journals.plos.org/plosone/s/submission-guidelines#loc-laboratory-protocols

We look forward to receiving your revised manuscript.

Kind regards,

Janhavi Ajit Vaingankar

Academic Editor

PLOS ONE

Journal Requirements:

4. Please provide additional details regarding participant consent. In the ethics statement in the Methods and online submission information, please ensure that you have specified (1) whether consent was informed and (2) what type you obtained (for instance, written or verbal). If your study included minors, state whether you obtained consent from parents or guardians. If the need for consent was waived by the ethics committee, please include this information.

Hui-Juan Cao is supported by the National Natural Science Foundation of China (81804000) and the Beijing Municipal Organization Department (2017000020124G292).

Reviewers' comments:

Reviewer's Responses to Questions

**Comments to the Author**

1. Is the manuscript technically sound, and do the data support the conclusions?

Reviewer #1: Partly

Reviewer #2: Yes

2. Has the statistical analysis been performed appropriately and rigorously? 

Reviewer #1: Yes

Reviewer #2: N/A

3. Have the authors made all data underlying the findings in their manuscript fully available?

Reviewer #1: Yes

Reviewer #2: Yes

4. Is the manuscript presented in an intelligible fashion and written in standard English?

Reviewer #1: Yes

Reviewer #2: Yes

5. Review Comments to the Author

Reviewer #1: 1.In the part of study design, 10 samples were described directly. It maybe suitable for the qualitative methods, therefore, it is needed to be revised.

2.Only participants who had participated in the study were interviewed, so the attitudes of those who were eligible but refused to participate and those who dropped out were also important factors.

3.The interview outline is not given in this paper, which needs to be supplemented.

4.To provide a new perspective, the new vision should be more prominent and specific description in the discussion, enhance the innovation of the article.

Reviewer #2: Dear authors:

It is great to read your work! There are several strengths of your manuscript. Especially I think you have chosen one important research topic. I also have some comments for your consideration:

1.Why only chose the person who had participated the trials before? Why had not chosen the person who rejected to participate the trials? Please justify

2. They are difference types of acupuncture trials. How the trials types affected the perceptions of participants?

3. What are the implications for recruitment for acupuncture trials in the future? Please clarify.

4. In INTRODUCTION, please clarify the current challenges in recruiting trial participant for acupuncture trials

5. In addition, your findings are specific to China?

Good luck!

6. PLOS authors have the option to publish the peer review history of their article (what does this mean?). If published, this will include your full peer review and any attached files.

Reviewer #1: No

Reviewer #2: No

---

## [Author Response · Author response to Decision Letter 0]

15 Feb 2020

Dear Editor Vaingankar,

Thank you for the opportunity to revise our manuscript; the comments and suggestions are very pertinent and helpful. We have revised the manuscript and responded to the comments point by point as outlined below, and we believe the manuscript is much improved as a result.

Please let us know if there are any further amendments you require for the manuscript.

We look forward to your reply.

Kindly regards,

Huijuan Cao and the author team

Additional comments:

1. Provide further background on compliance to acupuncture trial among Chinese and other races and why this is relevant in medicine

RE: We had previously conducted a literature review, summarising missing data in Chinese acupuncture clinical trials for the period 2006-2015, exploring the influence of missing data on study outcomes through secondary analysis. We have cited cite this paper as the first reference of this manuscript. In response to your comment, we have further added key results of that review in the Background section, to clarify the issue of patient compliance in acupuncture trials. 

2. Elaborate further on measures that were taken to reduce selection bias through refusals in this study. Given that this study set out to understand compliance related perspectives of clinical trial patients, by including only those who had good compliance and had repeatedly participated in other trials in the past, has several implications for this study. The issue of non-compliance may not have been adequately explored. Authors should comment on these in the discussion adequately.

RE: Thank you. As per our reply to the previous two reviewers who also raised this point, , we agree that our participant sample may not reflect the attitudes of those who did not have good compliance to acupuncture trials, and have added text to address this limitation, as outlined in our response to Reviewer 1, comment #2 above. We also suggested future studies to involve “both participants who withdrew from clinical trials prior to completion, and the acupuncturists or researchers directly involved in the trials, to better understand the factors influencing patient compliance from the perspective of all trial stakeholders”.

3. Elaborate on whether saturation was adequately achieved in this study. 30 of the 40 approached had refused participation. Were there other eligible patients that the study team could have approached had saturation not been reached. The 10 participants also seem to be quite heterogenous in their backgrounds. It is also not explained what type of trials they had participated in for example, whether there were of long/ short duration, included invasive procedures or were meant for specific health conditions. All these will have bearings on the qualitative content. Authors should elaborate on these points in the context of data / thematic saturation and implications of the findings.

RE: Sorry we did not make this clear. It was not that 30 of the 40 refused participation, but rather that we only contacted 10 of the 40. As we mentioned in the methods section, “Participants were continuously sampled from the convenience sample, with sample size determined according to the principle of information saturation. Recruitment was terminated when no new themes were derived from the interview data.” We contact them one by one, and the recording of the previous interviewee was transcribed and analyzed before we contact to the next person. “When the analysis of the 9th and 10th interview produced no new codes or concepts, we determined that data saturation had been reached, and no further participants were recruited.” We have added this sentence to the first paragraph of the Results section.

Journal Requirements:

RE: We have revised the manuscript according to the PLOS ONE style.

RE: We have included the captions of the Supporting Information files at the end of the manuscript, also we revised the in-text citations as S1 Table and S2 Table.

RE: Data cannot be shared publicly because of recordings of the conversations are private, which contain potentially identifying and personal information. Data are available from the Ethics Committee of Dong Zhi Men Hospital (contact via Dr.Yu) for researchers who meet the criteria for access to confidential data.

The data underlying the results presented in the study are available from Dr. Changhe Yu (yakno2@163.com), via below mail address:

Department of massage, Dong Zhi Men Hospital, 

No. 5 of Haiyuncang Road, Dongcheng District, 

Beijing, 100701

China

4. Please provide additional details regarding participant consent. In the ethics statement in the Methods and online submission information, please ensure that you have specified (1) whether consent was informed and (2) what type you obtained (for instance, written or verbal). If your study included minors, state whether you obtained consent from parents or guardians. If the need for consent was waived by the ethics committee, please include this information.

RE: We’ve mentioned in the Methods section that “Potential participants were informed about the researchers’ background, occupation and credentials. They were given an information sheet which described the purpose of the research, and advised by the researcher that they could withdraw from the study at any time, and request their audio recordings to be destroyed. Those in agreement then signed a written consent form prior to their interview taking place.”

Hui-Juan Cao is supported by the National Natural Science Foundation of China (81804000) and the Beijing Municipal Organization Department (2017000020124G292).

RE: Thank you! We removed the funding information from the manuscript. Please update the Funding Statement as “Hui-Juan Cao is supported by the National Natural Science Foundation of China (81804000) and the Beijing Municipal Organization Department (2017000020124G292). Jian-Ping Liu is supported by the Key project of the National Natural Science Foundation of China (No.81830115). The funders had no role in study design, data collection and analysis, decision to publish, or preparation of the manuscript.”

---

## [Decision Letter · Decision Letter 1]

1 Apr 2020

Factors influencing participant compliance in acupuncture trials: An in-depth interview study

PONE-D-19-30876R1

Dear Dr. Cao,

We are pleased to inform you that your manuscript has been judged scientifically suitable for publication and will be formally accepted for publication once it complies with all outstanding technical requirements.

With kind regards,

Janhavi Ajit Vaingankar

Academic Editor

PLOS ONE

Additional Editor Comments (optional):

Reviewers' comments:

Reviewer's Responses to Questions

**Comments to the Author**

1. If the authors have adequately addressed your comments raised in a previous round of review and you feel that this manuscript is now acceptable for publication, you may indicate that here to bypass the “Comments to the Author” section, enter your conflict of interest statement in the “Confidential to Editor” section, and submit your "Accept" recommendation.

Reviewer #1: All comments have been addressed

Reviewer #2: All comments have been addressed

2. Is the manuscript technically sound, and do the data support the conclusions?

Reviewer #1: Yes

Reviewer #2: Yes

3. Has the statistical analysis been performed appropriately and rigorously? 

Reviewer #1: Yes

Reviewer #2: N/A

4. Have the authors made all data underlying the findings in their manuscript fully available?

Reviewer #1: Yes

Reviewer #2: Yes

5. Is the manuscript presented in an intelligible fashion and written in standard English?

Reviewer #1: Yes

Reviewer #2: Yes

6. Review Comments to the Author

Reviewer #1: The author explains the comments in detail, and makes corresponding modifications and supplements to the contents of the article.Although in terms of interview samples, the patients who fell off or rejected to participant in the trials were not interviewed, which may lack some evidence to support the argument in the article, this paper can be drawn from the patients who have participated in the trial, which has certain clinical significance.

Reviewer #2: (No Response)

7. PLOS authors have the option to publish the peer review history of their article (what does this mean?). If published, this will include your full peer review and any attached files.

Reviewer #1: No

Reviewer #2: No

---

## [Editor Report · Acceptance letter]

3 Apr 2020

PONE-D-19-30876R1 

Factors influencing participant compliance in acupuncture trials: An in-depth interview study 

Dear Dr. Cao:

I am pleased to inform you that your manuscript has been deemed suitable for publication in PLOS ONE. Congratulations! Your manuscript is now with our production department. 

With kind regards,

on behalf of

Ms Janhavi Ajit Vaingankar 

Academic Editor

PLOS ONE